# Direct Anterior versus Lateral Approach for Femoral Neck Fracture: Role in COVID-19 Disease

**DOI:** 10.3390/jcm11164785

**Published:** 2022-08-16

**Authors:** Giuseppe Maccagnano, Francesco Maruccia, Michela Rauseo, Giovanni Noia, Michele Coviello, Andrea Laneve, Alessandro Pio Quitadamo, Giacomo Trivellin, Michele Malavolta, Vito Pesce

**Affiliations:** 1Orthopaedics Unit, Department of Clinical and Experimental Medicine, Faculty of Medicine and Surgery, University of Foggia, Policlinico Riuniti di Foggia, 71122 Foggia, Italy; 2Department of Anesthesia and Intensive Care, University of Foggia, Policlinico Riuniti di Foggia, 71122 Foggia, Italy; 3Orthopaedic & Trauma Unit, AOU Consorziale Policlinico. Department of Basic Medical Sciences, Neuroscience and Sense Organs, School of Medicine, University of Bari “Aldo Moro”, AOU Consorziale Policlinico. Piazza Giulio Cesare 11, 70124 Bari, Italy; 4Hip and Trauma Surgery Department, Piero Pederzoli Private Hospital, Peschiera del Garda, 37019 Verona, Italy; 5Knee Surgery Department, Piero Pederzoli Private Hospital, Peschiera del Garda, 37019 Verona, Italy

**Keywords:** COVID-19, proximal femoral fractures, direct lateral approach, direct anterior approach, DAA, hemiarthroplasty

## Abstract

Background: During the COVID-19 emergency, the incidence of fragility fractures in elderly patients remained unchanged. The management of these patients requires a multidisciplinary approach. The study aimed to assess the best surgical approach to treat COVID-19 patients with femoral neck fracture undergoing hemiarthroplasty (HA), comparing direct lateral (DL) versus direct anterior approach (DAA). Methods: A single-center, observational retrospective study including 50 patients affected by COVID-19 infection (30 males, 20 females) who underwent HA between April 2020 to April 2021 was performed. The patients were allocated into two groups according to the surgical approach used: lateral approach and anterior approach. For each patient, the data were recorded: age, sex, BMI, comorbidity, oxygen saturation (SpO_2_), fraction of the inspired oxygen (FiO_2_), type of ventilation invasive or non-invasive, HHb, P/F ratio (PaO_2_/FiO_2_), hemoglobin level the day of surgery and 1 day post operative, surgical time, Nottingham Hip Fractures Score (NHFS) and American Society of Anesthesiologists Score (ASA). The patients were observed from one hour before surgery until 48 h post-surgery of follow-up. The patients were stratified into five groups according to Alhazzani scores. A non-COVID-19 group of patients, as the control, was finally introduced. Results: A lateral position led to a better level of oxygenation (*p* < 0.01), compared to the supine anterior approach. We observed a better post-operative P/F ratio and a reduced need for invasive ventilation in patients lying in the lateral position. A statistically significant reduction in the surgical time emerged in patients treated with DAA (*p* < 0.01). Patients within the DAA group had a significantly lower blood loss compared to direct lateral approach. Conclusions: DL approach with lateral decubitus seems to preserved respiratory function in HA surgery. Thus, the lateral position may be associated with beneficial effects on gas exchange.

## 1. Introduction

The Severe Acute Respiratory Syndrome 2 (SARS-CoV-2), also known as COVID-19, is a new coronavirus identified for the first time at the end of 2019 in patients affected by pneumonia in the Wuhan region (China) [1].

The outbreak pandemic caused by SARS-CoV-2 strongly affected the World Health Organization. The COVID-19 syndrome is characterized by a broad spectrum of clinical presentations, ranging from the common cold to more serious syndromes, such as MERS (Middle East respiratory syndrome) and SARS (severe acute respiratory syndrome) [1,2]. An increased risk of mortality associated with COVID-19 infection was reported, and correlated with age, cardiovascular and chronic respiratory disease, diabetes mellitus, hypertension, obesity, black or South Asian identity, male sex and cancer [3,4]. Specifically, the elderly patients, the immunocompromised and those with pre-existing comorbidities present a higher risk of severe complications and death from COVID-19. Indeed, the patients subjected to orthopedic surgery for femoral neck fracture, registered a higher mortality rate [5,6,7].

Therefore, the management of these patients requires a multidisciplinary approach to improve the standards of care. Since December 2019, hospitals have implemented protocols to reorganize health care and manage the emergency. These changes led to a severe impact on surgical activity. During the first wave of the COVID-19 emergency, a reduction in major orthopedic trauma and activity-related trauma was observed, despite the incidence of fragility fractures in elderly patients remaining unchanged [8]. A multicenter study conducted in Italy during the “phase 1” (23 February–3 May 2020) showed a decrease in the surgical interventions for proximal femur fractures, but an increase in the domestic trauma [9]. Regarding the timing of surgery, the literature is also contradictory for orthopedic surgery [10]. According to some authors, early surgery may induce a second hit and cytokine stress, while usually early treatment lowers the risk of complications, such as DVT (deep vein thrombosis), UTI (urinary tract infections) and bedsores, as with non-COVID-19 patients [11]. A hip fracture is associated with a one-year mortality rate in aging patients ranging from 14% to 36%, showing a general reduction in the quality of life [12]. The patients with neck femur fracture may require internal fixation, hemiarthroplasty (HA) and total hip arthroplasty (THA), but the optimal treatment remains debated [13,14].

Hip replacement can be performed by the orthopedic surgeon, using different surgical approaches. The most common approaches are direct lateral, posterior-lateral in lateral position and anterior approach in supine position [15]. The direct anterior hip approach is considered a tissue sparing approach. Specifically, two intermuscular planes are involved: the superficial is identified between the sartorius muscle and the tensor fasciae latae, while the deep plane passes between the rectum femoral and tensor fascia latae. The direct lateral hip approach, according to Hardinge, involved the dissection of the fibers of the gluteus medius and vastus laterals to reach the joint. The recent findings are contradictory and the authors are not able to underline any superiority between the two procedures [16,17].

With respect of a hip fragility fracture, the one-year mortality ranges from 14% to 36%, with a general reduction in the quality of life [12]. In the previous studies, authors reported poor outcomes in patients affected by a fracture of the proximal femur and COVID-19 at admission [18]. Moreover, a multicenter cohort study reported an increased 30-day mortality for patients with COVID-19 infection requiring surgery [5]. Using different scores, it is possible to stratify the mortality risk at 30 days and functional outcomes for the patients affected by hip fracture; the Nottingham Hip Fracture Score (NHFS) is a valid index, according to the literature [19]. Due to respiratory system impairment in COVID-19 patients, the choice between the lateral or supine position could be challenging for the anesthesiologist. The non-COVID-19 patients treated with DAA compared to the lateral approach have shown shorter hospitalization and faster recovery during the first postoperative period. It is unclear whether DAA in patients with COVID-19 infection can be superior than lateral approach [20,21,22].

The aims of the current study are: (1) to assess the more appropriate surgical approach in COVID-19 patients undergoing HA, comparing direct lateral (DL) versus direct anterior approach (DAA); and (2) to evaluate the impact of the intraoperative position of the patient and how it may play a role on the respiratory function in a short-term time window.

## 2. Materials and Methods

This is a retrospective, case-control, mono-center study, validated by the Ethics Committee (protocol number: 11/CE/2022—31 December 2021).

The patients affected by a fragility femoral neck fracture were enrolled and treated at the Riuniti Hospital of Foggia between April 2020 and April 2021.

The patients were divided into two groups. Group A was represented by patients who underwent hip replacement by a lateral approach and group B by the patients treated by the anterior approach.

For each patient, the following data were recorded: age; sex; BMI; comorbidity; oxygen saturation (SpO_2_); fraction of the inspired oxygen administered (FiO_2_); type of ventilation (invasive or non-invasive); HHb; P/F ratio (PaO_2_/FiO_2_); hemoglobin level on the day of surgery and 1 day post-operative; surgical time; COVID-19 disease; Nottingham Hip Fractures Score (NHFS) [19] and American Society of Anesthesiologists Score (ASA) [23].

The data were recorded at the following times: T0 (one hour before the surgical procedure); T1 (one-hour post-surgery); T2 (24 h post-surgery); and T3 (48 h post-surgery).

The COVID-19 positivity was confirmed by a nose-pharyngeal molecular swab, followed by polymerase chain reaction technique (PCR), routinely designated for COVID-19 diagnosis [24] and a suggested chest X-ray [25].

Regarding the anesthesiology classification, the patients were stratified into five groups, according to Alhazzani et al. [26], as asymptomatic, mild, moderate, severe and critical illness.

The diagnosis of femoral neck fracture was detected through a pelvic X-Ray.

The inclusion criteria were:femoral neck fractures type 31.B according to the A.O. classification [27] and simultaneous COVID-19 infection at the time of the surgical procedure;age > 70 years;patients eligible for surgery.Exclusion criteria were:Parkinson’s disease;Hemoglobin < 8 g/dL;Patient underwent general anesthesia;ASA score 5.

The patients were surgically treated by two senior surgeons (G.M. and F.M.) with more than 10 years of hip surgery, who selected the appropriate surgical approach based on their respective surgical practice [28]. The lateral hip approach according to Hardinge [29] was used by G.M. in 25 cases (group A) and the direct anterior hip approach [30] was used by F.M in the other 25 patients (group B). In all of the cases, the same hip prosthetic implant was adopted.

Initially, we compared the variables in the study group (COVID-19 patients) in order to demonstrate the differences between the surgical approaches.

Afterwards, we analyzed the differences between the study and control group (non-COVID-19 patients), made up of fifty patients with the same fractures and treated by the two different approaches of the study groups.

The data were collected into a database, using Microsoft Excel software and analyzed using Software SPSS 25.0 (IBM, Armonk, NY, USA). A difference of 0.75 g/dL in the hemoglobin serum level was considered to be the minimal clinically important [31]. A power analysis, with a power of 80% and an alpha of 0.05, showed a sample size of fifteen patients per group.

The continuous variables were reported as the mean, one standard deviation and range [minimum–maximum], and the categorical variables were reported as number and percentage. Due to the non-homogeneous distribution of the values using the Kolmogorov–Smirnov test (*p* > 0.05), non-parametric tests were considered. To compare the average values between the groups at the same times, the U Mann–Whitney test or Fischer’s test were used, when appropriate. To compare the value within the same group at different times, the Wilcoxon test and Related-Samples Friedman’s test Two-Way Analysis of Variance were used. To demonstrate the correlation between the surgical approach and variables, the Spearman’s Rho correlation was used. A multiple regression model including the control group was then fitted for anemia and respiratory values in order to evaluate the effect of the COVID-19 disease compared to different approaches (“beta” coefficients). For all of the tests, a *p*-value of less than 0.05 was considered to be statistically significant.

Data availability: The data presented in this study are available on request from the corresponding author.

## 3. Results

Fifty consecutive patients with COVID-19 disease were enrolled in this study and allocated into two groups, 25 cases in each group. We compared the study and control group at recruitment. (Table 1). Six patients (12%) died 48 h after surgery, two in group A and four in group B (Table 2).

The hemoglobin serum levels before and after surgery are shown in Figure 1 for study group.

Most of the common comorbidities recorded were: 38 (76%) hypertension; 31 (62%) diabetes mellitus type II; 29 (58%) hypercholesterolemia; 17 (34%) gastroesophageal reflux disease; 14 (28%) hypothyroidism; 10 (20%) chronic obstructive pulmonary disease; 9 (18%) atrial fibrillation; 9 (18%) chronic kidney disease; 8 (16%) dementia; 3 (6%) inflammatory bowel disease; 11 (22%) with a history of neoplasia (4 rectum, 3 lung, 2 mammary, 1 kidney, 1 liver); 2 rheumatoid arthritis (4%) and 1 multiple sclerosis (2%). A total of 13 patients (26%) showed ≥4 comorbidities. The Nottingham Hip Score frequencies were: 3 points in 2 cases (4%); 4 points in 13 patients (26%); 5 in 22 (44%); 6 in 9 (18%); 7 in 3 (6%) and 8 points in 1 patient (2%). The patients were classified according to ASA score: 15 patients were ASA 2 (30%), 8 patients in group A and 7 patients in group B; 28 patients were ASA 3 (56%), 13 in group A and 15 in group B; and 7 were ASA 4 (14%), 4 cases in group A and 3 in group B.

Regarding respiratory assistance, at T0, the patients were ventilated as follows: 25 patients (50%) without oxygen support; 13 patients (26%) with Venturi’s Mask; 8 patients received noninvasive ventilation (NIV); 3 patients received CPAP and 1 patient received HFNC (High Flow Nasal Cannula). At T1, 13 patients (26%) changed their ventilation support, based on clinical decision (Table 3). According to Alhazzani et al. [26], the patients were divided into four groups, as shown in Table 3. None of the patients was classified as asymptomatic.

Regarding the P/F ratio during the follow-ups: at T0 the average value was 287 (77–619); at T1 it was 280 (79–595) at T2 it was 253 (53–514); at T3 it was 251.

The baseline P/F was 273 (77–486) and 302 (122–619) in groups A and B, respectively. Regarding group A, the P/F value increased to 297 (79–552) at T1, while for group B, the P/F value decreased to 263 (118–595) (Figure 2).

The average surgical time was 61 ± 8.47 min (47–77) for the lateral approach (group A) and 46.48 ± 4.72 min (38–56) for the direct anterior approach (group B) (*p* < 0.01) (Table 2). The mean hemoglobin loss was greater in group A (2.38 g/dL) than in group B (1.5 g/dL) at T1 (*p* < 0.01). In group A, 15 units of packet blood cells (PBC) were transfused, 6 units in group B within 48 h of surgery for each group (*p* < 0.01, Table 4).

At T1, all of the patients enrolled worsened. In fact, it emerged that the critical and severe classes increased, while the mild class reduced; furthermore, this tendency was statistically significant only for group B (*p* < 0.05) (Table 5 and Figure 3).

The patients classified as severe experienced more bleeding than the other groups over the reporting period (Table 6).

Table 7 reported the differences between the variables within each group.

At the end of follow-up (48 h), the mortality rate was 12% (six cases), four in group B (two ischemic myocardial attack, one pulmonary embolism and one pneumonia complications) and two in group A (one ischemic myocardial infarction and one pulmonary embolism).

A Rho Spearman’s correlation in the non-COVID-19 patients demonstrated no statistical differences between the two different approaches in the respiratory values but the anterior approach needed a lower PBC (coefficient = 0.16; *p* = 0.02).

The multiple linear regression models showed non-COVID-19 patients had higher Hb values. Moreover, all of the respiratory variables were influenced by COVID-19 rather than the different surgical approach (Table 8, Table 9 and Table 10).

We reported no statistical differences for the complications between the study and control group.

## 4. Discussion

In the current study, two different surgical approaches were used as treatment for femoral neck fractures: the direct lateral hip approach, according to Hardinge [29], and the direct anterior approach [30]. In relation to the aim of the study, the authors underlined the superiority of the DAA with respect to the DL in the COVID-19 patients with femoral neck fractures.

In the elderly patients with femoral neck fracture, the surgery should be performed within the first 24 h [32]. The mortality rate significantly increases if the operative treatment is delayed for more than 48 h [33]. On the other hand, Andritsos et al. [34], in a meta-analysis, showed how the timing of the surgery seems not to have any statistically significant impact on mortality in the COVID-19 patients with femoral neck fracture. Moreover, the conservative treatment in severely ill patients leads to issues with blood loss and increased bed rest, reducing the possibility of active patient management, in terms of pain and movement. The factors significantly associated with 30-day mortality were high ASA grade, older age, non-operative management, male sex, care/nursing home residence, a higher Nottingham Hip Fracture Score and a positive COVID-19 status [35]. The COVID-19 patients with multiple comorbidities (≥3 comorbidities) were associated with an increased risk of mortality [36,37,38]. The pre-existing comorbidities associated with the increased risk of complications were diabetes mellitus, hypertension and cardiorespiratory disease [39]. In our study 13 cases (26%) showed more than four pre-existing comorbidities and six of them (46.15%) died during the first 48 h. The pain relief achieved from surgery could justify the risks of peri-operative death, both for the patient at rest and during nursing care, even in the group at highest anesthetic risk [40].

Muñoz Vives JM et al. [41] have portrayed that 91.2% of patients infected with COVID 19 with a proximal femoral fracture underwent surgery with an ASA score lower than the conservative treatment group. Only 8.1% of the patients who underwent an operative treatment had ASA scores ≥ 4, while 50% of the patients managed non-operatively had ASA scores of ≥4, portraying a positive association between the type of treatment selected and the ASA score. In our study, we had surgically treated patients with an ASA score ≤ 4.

In a previous study, Pincus et al. [42] investigated the correlation between major surgical complications and different surgical approaches in patients treated for THA. Specifically, in a total of 30,098 surveyed patients, a higher significant risk of complications at 1-year follow-up was associated with the anterior approach. The difference, albeit statistically significant, was found to be mild (1 vs. 2%). The important biases were slightly different groups of patients operated on by different surgeons.

Conversely, Spina et al. [43] observed a greater benefit of the DAA compared with the DL approach in patients with Garden type III and IV femur fracture, in terms of blood loss, residual pain, functional recovery and mortality rate. Nevertheless, as shown by Moskal et al. [44], the complication rate is related to the learning curve [45]. In a meta-analysis conducted by Ramadanov et al. [16], the operation time of THA through the conventional lateral approaches was 17.8 min. shorter than the operation time of THA through DAA. On the other hand, in their meta-analyses, Wang et al. [46] and Higgins et al. [47] found no difference between DAA and conventional approaches in surgical times.

We observed a lower surgical time in the patients treated with DAA (61 min vs. 46.48 min) (*p* < 0.01) in the COVID-19 group. This was probably related to the different surgical theater set-up to prevent infection. In the non-COVID-19 group, no time difference was found between the two approaches. Several authors have shown longer surgical times for the anterior approach than the other approaches for both HA and THA [20,48].

There are several studies concerning the relationship between hip prosthesis and bleeding, recommending different protocols to manage the patient [49,50,51]. We adopted the same protocol for DAA and DL. To reduce the blood loss, 3 g of tranexamic acid were infused, including 2 g systemically and 1 g locally. A Redon drain was not positioned.

In their study, Parvizi et al. [52] analyzed 319 patients who were treated with THA, separating them into two groups, according to the surgical approach used (DAA vs. DL). The patients within the DAA group had a significantly lower blood loss compared to the treated group for the direct lateral approach. Similarly, in our study, we observed that the patients treated with DAA transfused six PBC units, while the DL group transfused 15 PBC units (*p* < 0.01). The major blood loss observed in the patients treated with HA for DL could be linked to the different anatomical structures involved during the surgical approach. In fact, for the DL approach, it is necessary to perform a dissection of the gluteus medius and vastus lateral and to sacrifice a muscular-tendinous flap of the minor gluteus, inducing bleeding. On the other hand for DAA, the surgeon has to identify an intermuscular and internervous plane, performing a tissue-muscle sparing.

The mortality rate for the positive COVID-19 patients was much higher than for non-infected patients, 30–35% and 7–10%, respectively [36,53,54]. In our study, the mortality rate was 12% during the first 48 h. We believe our low mortality rate may be due to the short follow-up period.

The Nottingham Hip Fracture Score (NHFS) in the patients with COVID-19 infection and in the patients who died at 48 h (two for each group), did not show a predictive value. Fell et al. [54] defined that NHFS was not reliable as a predictor for 30-day mortality in COVID-19 positive patients, and COVID-19 infection is an independent predictor for mortality in the neck fracture of femur patients, regardless of the NHFS. In a meta-analysis conducted by Kucukdurmaz et al. [55] and Miller et al. [56], there was no evidence to support the superiority of any approach, beyond a short follow-up in a non-COVID-19 population. Conversely, Putananon et al. [17] showed that the lateral approach has the best surgical approach for the higher HHS and lowest VAS pain after THA, followed by the anterior approach. Therefore, the choice of surgical approach should consider the experience and preference of the surgeon in a non-COVID-19 population. In contrast, the literature has not yet investigated the impact of the different hip approaches on COVID-19 patients.

The novelty of our study is the finding that the COVID-19 patients presenting with acute respiratory failure and undergoing the lateral approach for hip fracture, showed a better level of oxygenation when compared to the patients treated with the direct anterior approach (the pre-operative and day one post-operative P/F in the group A were 273.16 vs. 279.69, while the P/F in the group B were 301.56 vs. 226.72, *p* < 0.05, respectively, Table 7), despite the fact that the mortality of the COVID-19 patients remains high, due to the different immune-response and other independent factors related to the infection per se. These results are in line with the previous findings on improving oxygenation in lateral and prone positions [22,57,58,59,60,61,62]. Indeed, the previous data on COVID-19 patients showed that the acute respiratory failure COVID-19 patients responded well to proning and/or alternative body position [57,58,59,60,61,62], for several reasons arising from: (1) the redistribution of the ventilation to perfusion mismatch, due to the gravity-induced increase of the blood flow to the spared regions of the lung, which thus become better ventilated; and (2) the improved P/F ratio, due to the lung recruitment of previously dependent lung regions, that ameliorates the hypoxemic vasoconstriction, reducing the pulmonary vascular resistance and improving the right ventricular function. The need for surgery should exactly match the need for ventilator management of those patients, in order to avoid an increased risk of worsening the clinical condition after surgery and the need for high observation or a bed in ICU, given the lack of resources during the pandemic. We, thus, recommend clinicians to prefer the lateral approach over the direct anterior during the procedure in the mild and severe respiratory failure patients (or non-COVID-19), because patient recumbency is associated with a significant improvement in oxygenation and breathing pattern, with a good tolerance of the lateral position over the surgery time and no further recurrence of more invasive ventilatory strategy to counterbalance the effect of the prolonged supine position in patients with posterior decline bilateral infiltrates. Furthermore, we found no significant hemodynamic adverse effects in the lateral approach. The physiologic rationale for the lateral decubitus in non-intubated patients is strong, especially because, during the procedure, the patients are still able to continue or start a trial of NIV or HFNC, with no interruption in therapy and no need for intubation to control the upcoming worsening of breathing patterns (such as the increased work of breathing or dangerous minute ventilation). This can further deteriorate the acute respiratory failure or the requirement for prolonged invasive mechanical ventilation for several days, due to the deleterious effect of lying supine, with the injurious effect of a strong respiratory effort that can lead to excessive diaphragm activity with no limits on the tidal stress and strain imposed on the lungs and the onset of “patient self-inflicted lung injury” (P-SILI) [16,63].

We demonstrated an actual correlation between a post-operative P/F ratio and a reduced need for NIV or HFNC after surgery in the patients recumbent in the lateral position (Table 3), and this is probably due to P-SILI avoidance, together with a reduced requirement for the post-operative ICU, and to the protective role of the lateral positioning of patients with unilateral pleuro-parenchymal disease, with the normal lung down, which significantly affected the gas exchange. The experimental data in fact suggest that not all of the subjects are exposed to the development of P-SILI: patients with a P/F ratio below 200 mmHg may represent the most at-risk population, and the body positioning, together with NIV or HFNC, likely promotes the treatment success and may mitigate the lung injury. The unsuitable surgical approach choice for the type of patient could lead to complications, as emerged from our results, and therefore to greater public spending on patient care.

The results obtained from our surgical and anesthesiologic strategy are of great impact, and we believe that our results could be reproduced not only for future waves of COVID-19, but also for any kind of flu or viral pandemic in which acute respiratory failure dominates the scenario.

This study has some limitations. Firstly, the two senior surgeons have a different surgical practice in the hip surgery. This difference can influence the surgical time and the blood loss during the procedure. Secondly, the surgical time was manually measured by different operating room staff from the beginning of the surgical incision to the last stitch. Thirdly, the follow-up was limited and does not allow for the determination of the effects of the intra-operative position after the 48 h of data collection. On the other hand, the strong points of this work are the selection of the treated patients, respecting stringent inclusion and exclusion criteria and the volume of data collected and their analysis. Unlike other prospective or retrospective studies, different endpoints were studied, such as respiratory and blood outcomes. Further studies, with a longer follow-up, to understand the impact of body position on COVID patients with femoral neck fracture are needed. This could be one of the first studies comparing different surgical approaches for hip fracture and other widespread disorders, such as breast cancer or pneumonia, for example.

## 5. Conclusions

The choice of a surgical approach should consider the experience and preference of the surgeon in a non-COVID-19 patient with femoral neck fracture. The COVID-19 patients, treated with HA, showed preserved respiratory function mainly when a direct lateral approach with lateral decubitus was used, instead of the DAA with supine decubitus, despite the fact that a transient increase in blood loss was observed. Due to these results, the authors suggest the choice of the direct lateral approach in those COVID-19 patients classified as “moderate” or “severe”, according to Alhazzani. On the other hand, the “mild” patients may be treated according to the surgeon’s preference. The ASA score could be a good parameter for assessing the patient’s operability, when it is ≤4. The COVID-19 patients with multiple comorbidities (≥4 comorbidities) were associated with an increased risk of mortality and this should be considered in the decision-making process, but further studies are needed.

## Figures and Tables

**Figure 1 jcm-11-04785-f001:**
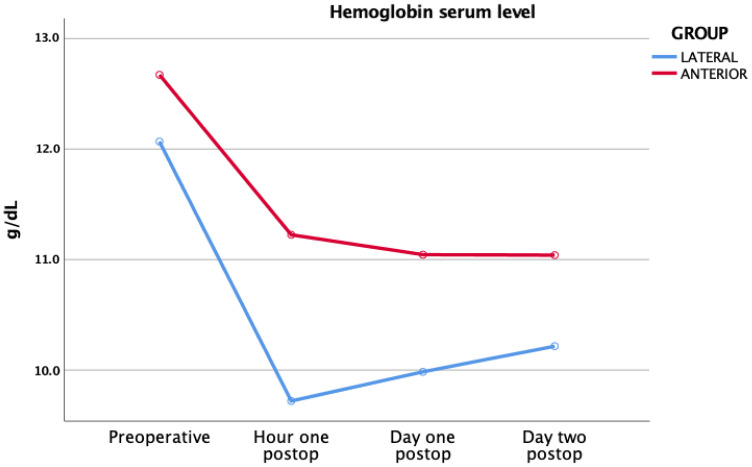
Hemoglobin serum levels within each group.

**Figure 2 jcm-11-04785-f002:**
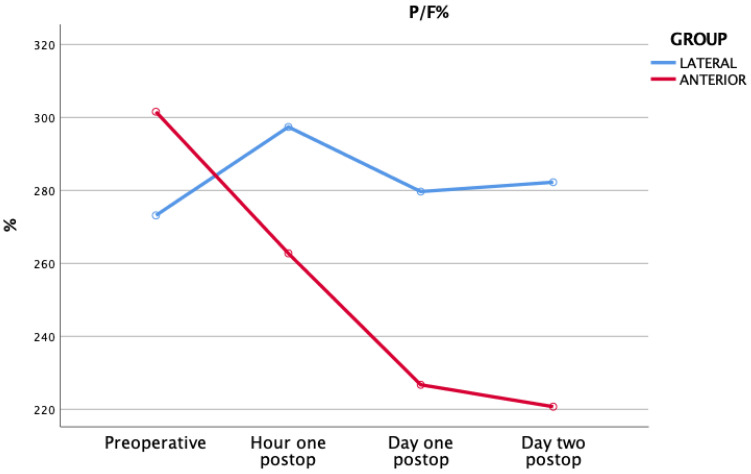
P/F within each study group.

**Figure 3 jcm-11-04785-f003:**
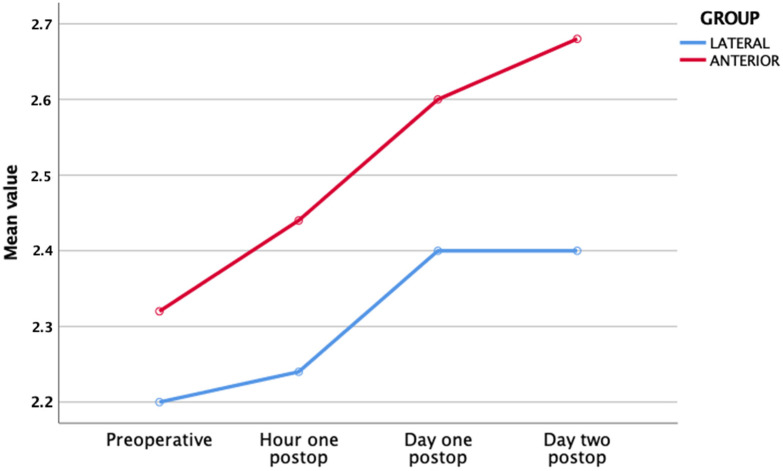
Alhazzani score mean value within each study group with differences between pre-operative and post-operative scores.

**Table 1 jcm-11-04785-t001:** Main data of the study (hundred patients, BMI: Body Mass Index).

Age	Study Group	Control Group	*p*-Value
Mean ± SD	77.02 ± 6.03	82.70 ± 4.89	**0.01**
Gender			
Female. *n* (%)	20 (40%)	33 (66%)	**0.01**
BMI (kg/cm^2^)			
Mean ± SD	26.59 ± 4.11	24.92 ± 2.32	**0.02**
Surgical Approach			
Lateral. *n* (%)	25 (50%)	25 (50%)	
Anterior. *n* (%)	25 (50%)	25 (50%)	
Surgical Time (min)			
Mean ± SD	53.74 ± 9.98	53.28 ± 0.09	0.84
Nottingham Hip Fracture Score			
Mean ± SD	5.02 ± 1.02	5.04 ± 1.01	0.87
ASA score			
2. *n* (%)	15 (30%)	13 (26%)	0.84
3. *n* (%)	28 (56%)	31 (62%)	
4. *n* (%)	7 (14%)	6 (12%)	

Statistically significant data in bold (*p* < 0.05).

**Table 2 jcm-11-04785-t002:** Pre-operative study group data divided by surgical approach (U Mann–Whitney test except for Fischer test for sex, HHb: hemoglobin subunit beta; BMI: Body Mass Index).

Mean ± SD or *n* (%)	Group A	Group B	*p*-Value
Age (year)	75.64 ± 5.13	78.40 ± 6.63	1.37
Sex (female)	10 (40%)	10 (40%)	0.61
BMI (Kg/cm^2^)	27.38 ± 3.97	25.82 ± 4.18	0.16
Surgical Time (min)	61 ± 8.47	46.48 ± 4.72	**0.01**
Nottingham Hip fracture Score	4.85 ± 0.99	5.20 ± 1.04	0.16
Pre-operative hemoglobin (g/dL)	12.07 ± 1.74	12.67 ± 1.82	0.27
Pre-operative pH	7.44 ± 0.08	7.42 ± 0.08	0.71
Pre-operative pO_2_ (mmHg)	88.4 ± 30.98	80.48 ± 25.65	0.21
Pre-operative pCO_2_ (mmHg)	40 ± 8.66	40.20 ± 9.17	0.92
Pre-operative HHb	2.50 ± 2.19	4.15 ± 3.29	0.08
Pre-operative P/F	273.16 ± 120.09	301.56 ± 147.33	0.76
Pre-operative Alhazzani score	2.20 ± 0.87	2.32 ± 0.85	0.60

Statistically significant data in bold (*p* < 0.05).

**Table 3 jcm-11-04785-t003:** Ventilation type and Alhazzani scores divided by Surgical Approach at different times for study group.

Different Times		Pre-Operative		One Hour Post-Operative		Day One Post-Operative		Day Two Post-Operative	
Ventilation Type	Group A/Group B	*n* (%)	*n* (%)	*n* (%)	*n* (%)	*n* (%)	*n* (%)	*n* (%)	*n* (%)
	No oxygen support	10 (40%)	15 (60%)	8 (32%)	7 (28%)	6 (24%)	5 (20%)	6 (24%)	5 (20%)
	Conventional oxygen therapy	0 (0%)	0 (0%)	2 (8%)	0 (0%)	3 (12%)	0 (0%)	3 (12%)	0 (0%)
	Venturi	8 (32%)	5 (20%)	8 (32%)	9 (36%)	9 (36%)	7 (28%)	10 (40%)	6 (24%)
	HFNC	1 (4%)	0 (0%)	1 (4%)	0 (0%)	1 (4%)	0 (0%)	4 (16%)	0 (0%)
	CPAP	0 (0%)	2 (8%)	0 (0%)	1 (4%)	0 (0%)	2 (8%)	0 (0%)	2 (8%)
	NIV	6 (24%)	3 (12%)	6 (24%)	8 (32%)	4 (16%)	11 (44%)	4 (16%)	12 (48%)
	Mechanical ventilation	0 (0%)	0 (0%)	0 (0%)	0 (0%)	2 (8%)	0 (0%)	2 (8%)	0 (0%)
Alhazzani scores									
	Critical	0 (0%)	0 (0%)	0 (0%)	0 (0%)	2 (8%)	10 (40%)	2 (8%)	2 (8%)
	Severe	12 (48%)	14 (56%)	12 (48%)	20 (80%)	11 (44%)	20 (80%)	11 (44%)	18 (72%)
	Moderate	6 (24%)	5 (20%)	7 (28%)	0 (0%)	7 (28%)	0 (0%)	7 (28%)	0 (0%)
	Mild	7 (28%)	6 (24%)	6 (24%)	5 (20%)	5 (20%)	5 (20%)	5 (20%)	5 (20%)

(HFNC: High Flow Nasal Cannula; CPAP: continuous positive airway pressure; NIV: noninvasive ventilation).

**Table 4 jcm-11-04785-t004:** Rho Spearman’s correlation between Surgical Time and Hemoglobin serum level at different times for study group.

Surgical Time	*p*-Value	Spearman’s Rank Correlation Coefficient
Pre-operative hemoglobin (g/dL)	0.39	−0.12
Hour One Post-operative hemoglobin (g/dL)	**0.02**	−0.33
Day One Post-operative hemoglobin (g/dL)	**0.04**	−0.29
Day Two Post-operative hemoglobin (g/dL)	0.08	−0.25

Statistically significant data in bold (*p* < 0.05).

**Table 5 jcm-11-04785-t005:** Differences between pre-operative and post-operative Alhazzani scores within each study group (Wilcoxon test).

	Different *p*-Value		
Pre-Operative vs.	One Hour Post-Operative	Day One Post-Operative	Day Two Post-Operative
Group A	0.56	0.06	0.06
Group B	0.18	**0.05**	**0.02**

Statistically significant data in bold (*p* < 0.05).

**Table 6 jcm-11-04785-t006:** Rho Spearman’s correlation between pre-operative Alhazzani score and Hemoglobin serum level at different times for study group.

	*p*-Value	Spearman’s Rank Correlation Coefficient
Pre-operative vs.	0.78	0.041
One hour Post-operative	0.34	0.136
Day One Post-operative	**0.03**	0.315
Day Two Post-operative	**0.02**	0.333

Statistically significant data in bold (*p* < 0.05).

**Table 7 jcm-11-04785-t007:** Differences between variables within each study group (Wilcoxon test and Related-Samples Friedman’s test Two-Way Analysis of Variance; HHb: hemoglobin subunit beta).

		Different Time *p*-Value			
		Pre-operative vs. Hour One	Hour One vs. Day One	Day One vs. Day Two	Related-Samples
Group A					
	Hb (g/dL)	**<0.01**	0.35	**0.01**	**<0.01**
	pH	0.70	0.81	0.57	0.95
	pO_2_ (mmHg)	0.22	0.08	0.47	0.05
	pCO_2_ (mmHg)	0.38	0.55	0.87	0.82
	HHb	0.69	0.08	0.05	0.09
	P/F	0.10	**0.03**	0.84	0.38
Group B					
	Hb (g/dL)	**<0.01**	**<0.01**	0.63	**<0.01**
	pH	0.80	0.50	0.71	0.90
	pO_2_ (mmHg)	0.05	**0.04**	0.06	0.17
	pCO_2_ (mmHg)	0.55	0.95	0.57	0.65
	HHb	**<0.01**	0.12	**<0.01**	**<0.01**
	P/F	**<0.01**	**<0.01**	0.14	**<0.01**

Statistically significant data in bold (*p* < 0.05).

**Table 8 jcm-11-04785-t008:** Multiple linear regression models for respiratory values (Fu: follow up; BMI: Body Mass Index).

	pH				pO_2_				pCO_2_			
	B	95% CI		*p*-Value	B	95% CI		*p*-Value	B	95% CI		*p*-Value
Intercept	7.51			**<0.01**	82.28			**<0.01**	47.13			**<0.01**
Non-COVID-19	−0.03	−0.04	−0.02	**<0.01**	−3.77	−7.28	−0.26	** 0.04 **	−3.40	−4.51	−1.57	**<0.01**
Anterior approach	−0.01	−0.02	0.01	0.65	−0.98	−4.64	2.67	0.60	−0.81	−2.34	0.72	0.30
Sex (female)	0.01	−0.01	0.02	0.35	−1.96	−5.33	1.43	0.26	−1.18	−2.60	0.23	0.10
Age	<0.01	−0.01	0.01	0.27	−0.37	−0.66	−0.08	** 0.01 **	−0.01	−0.13	0.12	0.95
BMI	−0.01	−0.01	0.01	0.08	1.19	0.67	1.71	**<0.01**	0.09	−0.13	0.31	0.42
Surgical Time	<0.01	0.01	0.01	0.10	0.12	−0.09	0.32	0.27	−0.07	−0.16	0.01	0.10
Fu time	<0.01	−0.01	0.01	0.70	−0.09	−1.41	1.24	0.90	0.57	0.22	1.13	0.05

Statistically significant data in bold (*p* < 0.05).

**Table 9 jcm-11-04785-t009:** Multiple linear regression models for respiratory values (HHb: hemoglobin subunit beta; Fu: follow up; BMI: Body Mass Index).

	HHb				P/F			
	B	95% CI		*p*-Value	B	95% CI		*p*-Value
Intercept	3.88			**<0.01**	505.29			**<0.01**
Non-Covid-19	−0.71	−1.28	−0.15	** 0.01 **	109.37	87.32	131.43	**<0.01**
Anterior approach	−0.21	−0.80	0.37	0.48	−13.57	−36.51	9.38	0.25
Sex (female)	0.48	−0.06	1.02	0.08	7.02	−14.19	28.24	0.56
Age	0.05	0.01	0.10	** 0.03 **	−4.92	−6.74	−3.10	**<0.01**
BMI	−0.20	−0.29	−0.12	**<0.01**	3.87	0.60	7.14	** 0.02 **
Surgical Time	0.02	−0.01	0.05	0.23	−0.77	−2.05	0.51	0.24
Fu time	−0.23	−0.44	−0.02	** 0.03 **	−7.56	−15.88	0.72	0.08

Statistically significant data in bold (*p* < 0.05).

**Table 10 jcm-11-04785-t010:** Multiple linear regression models for anemia (PBC: packet blood cells; Fu: follow up; BMI: Body Mass Index).

	Hb				PBC			
	B	95% CI		*p*-Value	B	95% CI		*p*-Value
Intercept	12.36			**<0.01**	−1.54			** 0.04 **
Non-COVID-19	−0.47	−0.73	−0.15	** 0.01 **	0.14	−0.01	0.30	0.08
Anterior approach	1.01	0.68	1.35	**<0.01**	−0.17	−0.33	−0.02	** 0.03 **
Sex (female)	−0.18	−0.49	0.13	0.26	−0.19	−0.34	−0.04	** 0.02 **
Age	−0.01	−0.04	0.02	0.54	0.01	−0.01	0.02	0.08
BMI	−0.04	−0.09	0.01	0.12	0.03	0.01	0.05	** 0.02 **
Surgical Time	−0.02	−0.03	0.01	0.13	0.01	0.01	0.02	** 0.01 **
Fu time	−0.14	−0.26	−0.02	** 0.02 **	−0.01	−0.06	0.06	0.98

Statistically significant data in bold (*p* < 0.05).

## Data Availability

The data presented in this study are available on request from the corresponding author. The data are not publicly available due to privacy.

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
