# Peer review of "Direct Anterior versus Lateral Approach for Femoral Neck Fracture: Role in COVID-19 Disease"

_jcm, 2022, doi:10.3390/jcm11164785_

Round 1

Reviewer 1 Report

in comparison to previously presented version I found he paper improved. Despite some minor inconsequences I believe the paper is worth publishing.

review 18.07.2022

[62] “…3rd2020) showed…” >> 3rd, 2020) showed

[82] In the Previous studies >> In the previous studies

[163] Kg >> kg

[170-173] „…Most common comorbidities recorded were: 38 (76%) hypertension, 31 (62%) diabetes mellitus type II, 29 (58%) hypercolesterolemia, 17 (34%) gastroesophageal reflux disease, 14 (28%) hypotiroidism, 10 (20%) chronic obstructive pulmonary disease,9 (18%) atrial fibrillation, 9 (18%) chronic kidney disease, 8 (16%) dementia, 3 (6%) infiammatory

[178] according by >> according to

[182] “…As regards the respiratory assistance,…” >> “…according to respiratory assistance,…”?

[188] asintomatic >> asymptomatic?

[185] HFNC [189 and 190] Hfnc . Authors should decide, whether they use the abbreviations HFNC or Hfnc and use them consequently in the whole text. The same remark relates to NIV >> Niv

[222] “…myocardial attack…” >> “heart attack” or “myocardial infarction” sound better

[227] Covid disease >> SARS-covid?

[251] were Diabetes >> were diabetes

[282-3] “…To reduce blood loss were infused 3 g of tranexamic acid (2g systemic and 1g local) and no redon was positioned…”  >> “…To reduce blood loss 3 g of tranexamic acid were infused, including 2 g systemically  and 1 g locally. Redon drain was not positioned….”

[299] Covid19 >> Covid-19

Authors should decide, whether they use dots in numbers (1.73; 2.34) as in tables 1-7 or commas (1,73; 2,34) as in tables 8-9.

I would also suggest validation of the text by native English-speaking person.

Author Response

In comparison to previously presented version I found he paper improved. Despite some minor inconsequences I believe the paper is worth publishing.

We thank the Reviewer for advice.

review 18.07.2022

[62] “…3rd2020) showed…” >> 3rd, 2020) showed

[82] In the Previous studies >> In the previous studies

[163] Kg >> kg

[170-173] „…Most common comorbidities recorded were: 38 (76%) hypertension, 31 (62%) diabetes mellitus type II, 29 (58%) hypercolesterolemia, 17 (34%) gastroesophageal reflux disease, 14 (28%) hypotiroidism, 10 (20%) chronic obstructive pulmonary disease,9 (18%) atrial fibrillation, 9 (18%) chronic kidney disease, 8 (16%) dementia, 3 (6%) infiammatory

[178] according by >> according to

[182] “…As regards the respiratory assistance,…” >> “…according to respiratory assistance,…”?

[188] asintomatic >> asymptomatic?

We corrected these errors.

[185] HFNC [189 and 190] Hfnc . Authors should decide, whether they use the abbreviations HFNC or Hfnc and use them consequently in the whole text. The same remark relates to NIV >> Niv

We appreciate your suggestion and decided to use capitalized abbreviations in the whole text.

[222] “…myocardial attack…” >> “heart attack” or “myocardial infarction” sound better

[227] Covid disease >> SARS-covid?

[251] were Diabetes >> were diabetes

[282-3] “…To reduce blood loss were infused 3 g of tranexamic acid (2g systemic and 1g local) and no redon was positioned…”  >> “…To reduce blood loss 3 g of tranexamic acid were infused, including 2 g systemically  and 1 g locally. Redon drain was not positioned….”

[299] Covid19 >> Covid-19

We corrected these errors.

Authors should decide, whether they use dots in numbers (1.73; 2.34) as in tables 1-7 or commas (1,73; 2,34) as in tables 8-9.

Thanks a lot, we decided to use dots in numbers in all the tables.

I would also suggest validation of the text by native English-speaking person.

Thanks a lot. We submitted the manuscript to a native English speaker who made several changes.

Reviewer 2 Report

1. First of all: English language is bad. You need to correct the whole manuscript by a native English speaker. I, myself, am not a native English speaker and I still saw the obvious problem with English language throughout the whole manuscript. I began to correct mistakes in English language and stopped in the methods section because it was too much.

2. In the title you use the expression “neck femoral fractures”. Throughout the manuscript you use other similar expressions like femur neck fracture. Please change them all to “femoral neck fracture”.

Femoral neck fracture instead of neck femoral fractures

3. Abstract: „The study aimed to assess the best surgical approach to treat Covid-19 patients with  proximal femoral fracture undergoing hemiarthroplasty (HA), comparing direct lateral (DL) versus direct anterior approach (DAA).“

I leave to the editor to decide if it is of interest to investigate surgical approaches to the hip joint in COVID patients. Why not in patients with breast cancer or pneumonia for example? 

4. Line 19: Here you say “proximal femoral fracture”. You mean “femoral neck fracture” – this is not the same, there a lot of types of proximal femoral fractures, one type is the femoral neck fracture. Please be more precise.

5. Line 23: Stick to uniform expressions - not “access” but “approach”. The same problem exists throughout the whole manuscript.

6. Line 26: Post operative - one word.

7. Line 29: It is “according TO” not “according BY”. This concerns the whole manuscript.

8. Line 29: “group”, not “group”.

9. Line 35: “seems to preserve” not “preserved”

10. Introduction: Line 42: “in patientS affected by” 

11. Line 53: „Indeed, patients subjected to orthopedic surgery for femoral neck fracture, have registered a mortality [5-7].“ I do not understand the meaning of this sentence here.

12. Line 64: “Authors” small letter. This concerns the whole manuscript.

13. Line 68: better: “elderly patients” 

14. Line 69: Now: “neck femur fracture”?

15. Line 75: “intermuscular planes” not “internervous” 

16. Line 78: “according TO Hardinge”

17. Line 80: “Authors” small again

18. Line 79-80: „According to the recent literature Authors are not able to underline the superiority between the two procedures [16].“ I disagree: 1. You reference Higgins who compared DAA to posterior approach in THA. He did not compare DAA to lateral approach. 2. There are meta-analyses that show better results for DAA when compared to conventional approaches including lateral approaches, for example: doi: 10.1038/s41598-021-00405-4. Please cite this article and change the sentence in: “literature is contradictory”.

19. Line 82 and 83: “Previous” and “Authors” small letters

20. Methods: Line 115: “polymerase chain reaction tecHnique”

21. “As regard”: as regard TO, the TO is missing throughout the whole manuscript. Better expression: “Regarding …”

22. Line 136: a prospective study!? Is it retrospective or prospective?

I think I understand what you did: you compared THA through DAA and lateral approach (groups A and B) retrospectively and then you compared those groups together (experimental group) prospectively with a non covid patient cohort called control group prospectively. If that is what you did you have to explain it better. It is hard to understand from your methods. Furthermore, I am not sure if this is methodologically correct.

23. “Surgical“ small letter. There is a problem with capitalization in the whole manuscript.

24. You should discuss your results with some of those systematic reviews and meta-analyses:

https://doi.org/10.1038/s41598-021-00405-4

https://doi.org/10.1016/j.arth.2014.10.020

https://doi.org/10.1177/1120700018820652

https://doi.org/10.1016/j.surge.2018.09.001

https://doi.org/10.1016/j.arth.2017.11.011

https://doi.org/10.1007/s00590-017-2046-1

https://doi.org/10.1186/s13018-018-0929-4

https://doi.org/10.1097/MD.0000000000002126

https://doi.org/10.1302/2058-5241.3.180023

https://doi.org/10.1302/0301-620X.99B6.38053

Author Response

1. First of all: English language is bad. You need to correct the whole manuscript by a native English speaker. I, myself, am not a native English speaker and I still saw the obvious problem with English language throughout the whole manuscript. I began to correct mistakes in English language and stopped in the methods section because it was too much.

Thanks a lot. We submitted the manuscript to a native English speaker who made several changes.

2. In the title you use the expression “neck femoral fractures”. Throughout the manuscript you use other similar expressions like femur neck fracture. Please change them all to “femoral neck fracture”.

Femoral neck fracture instead of neck femoral fractures

We corrected according to your suggestion.

3. Abstract: „The study aimed to assess the best surgical approach to treat Covid-19 patients with proximal femoral fracture undergoing hemiarthroplasty (HA), comparing direct lateral (DL) versus direct anterior approach (DAA).“

I leave to the editor to decide if it is of interest to investigate surgical approaches to the hip joint in COVID patients. Why not in patients with breast cancer or pneumonia for example?

We thank the Reviewer for the suggestion. However this study and this procedure were focus on Covid-19 patients. Future perspectives will allow the application of such techniques also in other type of ARF (acute respiratory failure) patients, using the same principle. Future trials are needed to investigate whether this approach could be useful to treat every patient with ARF.

4. Line 19: Here you say “proximal femoral fracture”. You mean “femoral neck fracture” – this is not the same, there a lot of types of proximal femoral fractures, one type is the femoral neck fracture. Please be more precise. 

We corrected according to your suggestion.

5. Line 23: Stick to uniform expressions - not “access” but “approach”. The same problem exists throughout the whole manuscript.

We corrected “approach” throughout the whole manuscript.

6. Line 26: Post operative - one word.

7. Line 29: It is “according TO” not “according BY”. This concerns the whole manuscript.

8. Line 29: “group”, not “group”.

9. Line 35: “seems to preserve” not “preserved”

10. Introduction: Line 42: “in patientS affected by”

11. Line 53: „Indeed, patients subjected to orthopedic surgery for femoral neck fracture, have registered a mortality [5-7].“ I do not understand the meaning of this sentence here.

12. Line 64: “Authors” small letter. This concerns the whole manuscript.

13. Line 68: better: “elderly patients”

14. Line 69: Now: “neck femur fracture”?

15. Line 75: “intermuscular planes” not “internervous”

16. Line 78: “according TO Hardinge”

17. Line 80: “Authors” small again

We corrected these errors.

18. Line 79-80: „According to the recent literature Authors are not able to underline the superiority between the two procedures [16].“ I disagree: 1. You reference Higgins who compared DAA to posterior approach in THA. He did not compare DAA to lateral approach. 2. There are meta-analyses that show better results for DAA when compared to conventional approaches including lateral approaches, for example: doi: 10.1038/s41598-021-00405-4. Please cite this article and change the sentence in: “literature is contradictory”.

Thank you very much for your suggestions.  We cited “: doi: 10.1038/s41598-021-00405-4”  and changed the sentence as follow: 

The recent literature is contradictory, authors are not able to underline the superiority between the two procedures [16,60].

Finally, we replaced Higgin's work with DOI: 10.1007/s00590-017-2046-1, which seemed more appropriate to our arguments.

19. Line 82 and 83: “Previous” and “Authors” small letters

20. Methods: Line 115: “polymerase chain reaction tecHnique”

21. “As regard”: as regard TO, the TO is missing throughout the whole manuscript. Better expression: “Regarding …”

We corrected these errors.

22. Line 136: a prospective study!? Is it retrospective or prospective?

I think I understand what you did: you compared THA through DAA and lateral approach (groups A and B) retrospectively and then you compared those groups together (experimental group) prospectively with a non covid patient cohort called control group prospectively. If that is what you did you have to explain it better. It is hard to understand from your methods. Furthermore, I am not sure if this is methodologically correct.

We better explained our study type.

This is a retrospective, case-control, mono-center study, validated by the ethics committee (protocol number: 11/CE/2022 - 31.12.2021)…

Initially, we compared the variables in the study group (Covid-19 patients) in order to demonstrate the surgical approaches differences.

Afterwards, we analyzed the differences between study and control group (non-Covid-19 patients), made up of fifty patients with the same fractures and treated by the same two different approaches of the study groups.

23. “Surgical“ small letter. There is a problem with capitalization in the whole manuscript.

We corrected these errors.

24. You should discuss your results with some of those systematic reviews and meta-analyses:

https://doi.org/10.1038/s41598-021-00405-4

https://doi.org/10.1016/j.arth.2014.10.020

https://doi.org/10.1177/1120700018820652

https://doi.org/10.1016/j.surge.2018.09.001

https://doi.org/10.1016/j.arth.2017.11.011

https://doi.org/10.1007/s00590-017-2046-1

https://doi.org/10.1186/s13018-018-0929-4

https://doi.org/10.1097/MD.0000000000002126

https://doi.org/10.1302/2058-5241.3.180023

https://doi.org/10.1302/0301-620X.99B6.38053

Thanks for your suggestions: we have discussed our results with some of those systematic reviews and meta-analyses:

https://doi.org/10.1038/s41598-021-00405-4

https://doi.org/10.1016/j.arth.2014.10.020

https://doi.org/10.1186/s13018-018-0929-4

https://doi.org/10.1016/j.surge.2018.09.001

https://doi.org/10.1016/j.arth.2017.11.011

https://doi.org/10.1007/s00590-017-2046-1

In a meta-analyis conducted by Ramadanov et al. [60], the operation time of THA through conventional lateral approaches was 17.8 min. shorter than the operation time of THA through DAA. On the other hand, in their meta-analyses Wang et al. [62] and Higgins et al. [16] found no difference between DAA and conventional approaches in surgical times.

atients with the same fractures and treated by the same two different approaches of the study groups…. In a meta-analysis conducted by Kucukdurmaz et al. [63] and Miller et al. [64], there were no evidence to support the superiority of any approach beyond a short follow-up in a non-Covid population. Conversely, Putanon et al. [61] showed that the lateral approach has the best surgical approach for the higher HHS and lowest VAS pain after THA, fol-lowed by the anterior approach. Therefore, the choice of surgical approach should con-sider experience and preference of the surgeon in a non-Covid population. In contrast, the literature has not yet investigated the impact of the different hip approaches on covid patients.

Reviewer 3 Report

These are my suggestions:

Generally, Improve the scientific soundness: use past tense, third person, passive voice

1.Title. making the title short and concise.

Discussion:

-Please, provide what is the evidence in the past and why is this study needed in comparison to the previous evidence?

- Please further discuss the details of your findings

- What are the clinical applications of your results? How they can improve the clinical practice?

- The meaning of the study: possible mechanisms and implications for clinicians or policymakers

- Strengths and weaknesses in relation to other studies, discussing particularly any differences in results

- Limitations of your study

- Unanswered questions and future research

Author Response

Generally, Improve the scientific soundness: use past tense, third person, passive voice

We revised the whole manuscript according to your suggestion

1.Title. making the title short and concise.

We changed title as follow:

Direct anterior versus lateral approach for femoral neck fracture: role in Covid-19 disease

Discussion:

-Please, provide what is the evidence in the past and why is this study needed in comparison to the previous evidence?

- Please further discuss the details of your findings

We completely revised the discussion section improving references, comparing our results with recent literature.

In a meta-analyis conducted by Ramadanov et al. [60], the operation time of THA through conventional lateral approaches was 17.8 min. shorter than the operation time of THA through DAA. On the other hand, in their meta-analyses Wang et al. [62] and Higgins et al. [16] found no difference between DAA and conventional approaches in surgical times.

Patients with the same fractures and treated by the same two different approaches of the study groups…. In a meta-analysis conducted by Kucukdurmaz et al. [63] and Miller et al. [64], there were no evidence to support the superiority of any approach beyond a short follow-up in a non-Covid population. Conversely, Putanon et al. [61] showed that the lateral approach has the best surgical approach for the higher HHS and lowest VAS pain after THA, fol-lowed by the anterior approach. Therefore, the choice of surgical approach should con-sider experience and preference of the surgeon in a non-Covid population. In contrast, the literature has not yet investigated the impact of the different hip approaches on covid patients.

- What are the clinical applications of your results? How they can improve the clinical practice?

- The meaning of the study: possible mechanisms and implications for clinicians or policymakers

We focused possible clinical mechanisms and implications for clinicians or policymakers in discussion and conclusion section.

…Actually, we demonstrated a correlation between a post-operative P/F ratio and a reduced need for NIV or HFNC after surgery in patient’s recumbent on the lateral position (Table 3), and this is probably due to P-SILI avoidance, together with a reduced re-quirement of post-operative ICU, and to the protective role of lateral positioning of pa-tients with unilateral pleuro-parenchymal disease with the normal lung down, which significantly affected gas exchange. Experimental data in fact suggest that not all subjects are exposed to the development of P-SILI: patients with a P/F ratio below 200 mmHg may represent the most at risk population, and that body positioning together with NIV or HFNC likely promotes treatment success and may mitigate lung injury. The unsuitable surgical approach choice for the type of patient could lead to complications, as emerged from our results and therefore to greater public spending on patient care.

…The choice of surgical approach should consider experience and preference of the surgeon in a non-Covid patient with femoral neck fracture. Covid-19 patients, treated with HA, showed preserved respiratory function mainly when a direct lateral approach with lateral decubitus was used instead of the DAA with supine decubitus, despite a transient increase of blood loss was observed. Due to these results, the authors suggest to choose the direct lateral approach in Covid-19 patients classified as “moderate” or “severe” according to Alhazzani. On the other hand, “mild" patients may be treated accordingly to surgeon's preference. ASA score could be a good parameter for assessing a patient's operability, when it is ≤ 4. Covid-19 patients with multiple comorbidities (≥4 comorbidities) were associated with an increased risk of mortality and this should be considered in the decision making process, but further studies are needed.

- Strengths and weaknesses in relation to other studies, discussing particularly any differences in results

- Limitations of your study

- Unanswered questions and future research

We improved discussion section with strengths, limitations, unanswered questions and future research according to your suggestions.

This study has some limitations. Firstly, the two senior surgeons have a different surgical practice in the hip surgery. This difference can influence the surgical time and the blood loss during the procedure. Secondly, the surgical time was manually measured by different operating room staff from the beginning of the surgical incision to the last stitch. Thirdly, the follow-up is limited and does not allow to determine the effects of the intraoperative position after the 48 hours of data collection. On the other hand, strong points of this work are the selection of treated patients, respecting stringent inclusion and exclusion criteria and the volume of data collected and their analysis. Unlike other prospective or retrospective studies, different endpoints have been studied, such as respiratory and blood outcomes. Further studies with a longer follow-up to understand the impact of body position in Covid patients with femoral neck fracture are needed. This could be one of the first studies comparing different surgical approaches for hip fracture and other widespread disorders as breast cancer or pneumonia for example.

Round 2

Reviewer 2 Report

Thank you for changing according to my suggestions. 

This manuscript is a resubmission of an earlier submission. The following is a list of the peer review reports and author responses from that submission.

Round 1

Reviewer 1 Report

Authors assessed differences in outcomes between Covid-19 patients undergoing hemiarthroplasty by comparing direct lateral and direct anterior approach. They found, that lateral approach (group A) led to a better oxygenation level but anterior approach (group B) led to lower blood loss.

I have some doubts regarding statistical methods. To compare differences in selected clinical parameters within the same group in four time points (T0, T1, T2, T3) a Related-Samples Friedman's test Two-Way Analysis of Variance should be used, instead of Wilcoxon-signed rank test.

Lines 152-155 duplicates what is already in the Table 1. Lines 160-169 I recommend to put this information into a table, rather than to a long text.

Line 160 please remove HTN.

Lines 177-181, 184, 185, please specify the values in the brackets []. Are they ranges or IQR?

Table 5 with correlations is missing. Table 4 is shown twice instead.

Table 3 and Table 6 could be joint together; so that the times could be as columns.

Figures: it is not clear to me what exactly has been plotted here and to what relate p-value(s). Did authors assess differences in hemoglobin (Fig.1), P/F (Fig.2), Alhzzani score (Fig.3) between lateral and anterior approach for each time point separately (by U-Mann Whitney test)?  If yes, then correction for multiple testing should be applied. Further, it is not clear to which time point relate p-value(s).

Reviewer 2 Report

Before any further analysis Authors should improve the text.  Unfortunately, due to linguistic reasons, some editorial mistakes and faults, i.e. cited several times author’s name Alhzzani instead of Alhazzani; missing explanation of HBB abbreviation used in the text; inappropriate name of the Author of statistical test written with small letter „spearman” instead of Spearman [204], and several others, the text is not intelligible. It also seems that punctuation marks (commas, full stops) are put in the text randomly.

In lines 52, 95,209, 212, 267, 270 and 275 Authors use unknown to me term „…medial fracture of the femur…”. Do they mean „…fracture of the proximal femur…”?

The sentence „…According to some authors who speak about second hit and cytokine stress after surgery, on the other hand early treatment prevents the risk of complications as deep vein thrombosis,  urinary tract infections and bedsores as non-COVID patients [11]…” is hardly understood. I would propose to divide it in 2-3 less complicated ones. I would also encourage Authors to re-evaluate the meaning of this sentence.

Authors use the term SatO2, but for me an alternative SpO2 seems to be far more often used.

The term „…oxygen level administered (FiO2)…” [line 101] could better be explained as „fraction of the inspired oxygen” originating from the FiO2 definition.

Lines 108-110:  In the sentence: „…The Covid-19 positivity was confirmed by a nose-pharyngeal molecular swab, followed by nucleic acid amplification technology for more accurate virus detection [20] and a suggestive chest X-ray [21]….” Authors seem to suggest that they have implemented very sophisticated molecular / genetic diagnosis of the COViD-19, including several nucleic acid amplification techniques, namely polymerase chain reaction, ligase chain reaction, strand displacement amplification, nucleic acid sequence-based amplification, transcription-mediated amplification, branched DNA, hybrid capture, DNA cleavage-based signal amplification, rolling circle amplification, and others. Nevertheless, I suppose, they have focused on standard polymerase chain reaction technique (PCR) routinely designated for COVID-19 diagnosis. 

Line 207: “…Test di Wilcoxon…” – Wilcoxon test

I would also propose to pay more attention on proper use of the commonly accepted abbreviations, like pH instead of PH and vs instead of VS (table 9).

Line 232 “…treated with DAA transferred 6 PBC…” – do Authors mean transfused 6 PBC?

Line 240 – numbers of cited publications (36-38) should be given in square brackets.

Lines 246-8: “…As regards the surgical treatment and approach, our data showed that the lateral position, when compared to the direct anterior approach, led to a better level of oxygenation (Group A T0 273.16-T1 279.69 vs Group B: T0 301.56-T2 226.72) (p< 0.05) (Graph 2)…” is not easily understood. As I suppose, Authors mean that lateral surgical approach (not position) is beneficial to COVID-19 patients due to less-disturbed ventilation (supposing - in obese patients?). This finding is of value, since it gives a valuable recommendation for patient’s positioning during surgical procedures in mild and severe respiratory insufficiency in COVID-19-positive patients. I would suggest to analyze / discuss this observation more profoundly.

Line 250: “…or alternative body potions…” – should be positions or fluids?

Line 251: “…reasons going from…” I have a feeling that coming would sound better.

Line 255: “…better post-operative P/F ratio…” – which P/F ratio is better?

Line : “…Contrariwise,..” is probably not the best word for this sentence

Meteorically

I do not understand, why Authors, excluding from the study patients with ASA score 5 (Materials and Methods) [124] had not operated patients with ASA score 1 and 4 [Table 1]? Surprisingly, some COVID-19-positive patients with femoral neck fracture could also be ASA-1.

Both analyzed groups seem not to be comparable according to Alhazzani score (Fig.3). Preoperative mean score for DAA group was higher than DL group. Were those differences significant or not?

Authors should also present and discuss, why patients with several respiratory insufficiency according to Alhazzani scale were operated due to fracture of the femoral neck? What was the reason and recommendation to perform hemiarthroplasy for severely ill patients (26 out from 50 were classified preoperatively as severe – Table 6). Additionally, do Authors believe that there were recommendations to perform hemiarthroplasty in COVID-ill patient additionally suffering from severe comorbidities, including dementia (3 patients), obstructive pulmonary (9), inflammatory bowel (14) and chronic kidney (8) diseases and multiple sclerosis (1)? This seems to me controversial analyzing the data presented on Figure 2, showing respiratory insufficiency in all patients included into the study. Preoperative mean (as might be supposed from this figure, but not specified) P/F value was ca 300 in DAA and ca 275 in DL groups, which could be stated as hypoxia for the DAA and respiratory failure for the DL group, dropping down to severe (P/F ca 225) in DAA group, postoperatively. I would like to encourage Authors to discuss this fact to valid both recommendations to the performed procedure and its results. Are there accessible data showing the postoperative mortality rate and complications of the performed procedure? [asking this question I am not willing to question performed orthopedic’ treatment, but I would like to validate complications that come from the COVID itself].

Table 1 shows very low percentage of female patients with femoral neck fracture (40%). This fact has not been analyzed in Discussion.

Tables 4 and 5 shows some inconsequence in the presented data. Day One Postoperative hemoglobin is shown to have p value between group 0,01:

Day One Postoperative hemoglobin (g/dL)

7.43±0.06

7.42±0.06

<0.01

And in Table 5 - 0,04:

Day One Postoperative hemoglobin (g/dL)

7.43±0.06

7.42±0.06

0.04

An analysis of the presented postoperative hemoglobin concentration seems not to be relevant, since Authors stated transfusion of blood (6 units in DAA group and 15 in DL group).

Round 2

Reviewer 1 Report

Dear Authors, 

thank you for providing the revised version of your manuscript. You however did not address my main points regarding statistics. 

Next, the correlation table does not include any correlation - correlation coefficient could be in ranges between minus 1 and plus one. You report however mean and SD. 

Reviewer 2 Report

There are some discrepancies in presented data. Once the Authors state that:

[168] „…Four patients died 48 hours after surgery, 2 for each group. (Table 1 and Table 2)…”. That means 4 from 50 = 8%.

And later on that:

[248-250] “…At the end of follow-up (48 hours), the mortality rate was 12 % (6 cases), 4 in Group B (2 ischemic myocardial attack, 1 polmunary embolism and 1 pneumonia complications) and 2 in group A (1 ischemic myocardial attack and 1 polmunary embolism)…”.

So, what was the mortality rate in this study during the first 48 postoperative hours?

By the way, please correct the world “polmunary” with “pulmonary” in lines 249 and 250.

Authors stated that [lines 257-9] “…The Hemiarthroplasty was considered as treatment option. As shown by Dan-Feng Xu et al [39] (doi: 10.1186/s13018-017-0528-9), the surgical option was associated with a higher union rate and a tendency toward less avascular necrosis than conservative treatment…” seems not to correspond to the cited reference [Xu, Dan-Feng et al. “A systematic review of undisplaced femoral neck fracture treatments for patients over 65 years of age, with a focus on union rates and avascular necrosis.” Journal of orthopaedic surgery and research vol. 12,1 28. 10 Feb. 2017, doi:10.1186/s13018-017-0528-9], which deals with femoral neck fractures that were stabilized, not replaced with hip prosthesis.  

In the presented paper the final outcome of operative orthopedic treatment does not depend on the healing of the fracture. More appropriate would be the publication on the same or corresponding problem that is analyzed in the current paper.

What do Authors mean by “..durable blood loss…”? [line 261]

The  sentence “…Scudiero et al [40], have demonstrated pulmonary embolism (PE) is a relatively common complication in COVID-19 and it is associated with increased mortality risk. Conservative treatment of femoral 265 neck fractures remains a valid option in high anesthetic risk, low demand patients...” [lines 263-7] is unarguably true, but do not correspond with the subject that is analyzed in the paper.

Later on [268-72] in the sentence “… The mortality rate was similar at one years, but higher in the initial period in non-operated patients. Surgery should be considered due to increase the possibility of movement with a decrease of pain and mortality. The pain relief achieved from surgery, could justify the risks of perioperative death, both for the patient at rest and during nursing care, even in the group at highest anesthetic risk [42]…” Authors ignored methodological facts given by Authors of cited publication, which clearly explains that “…Patients were managed non-operatively if they were felt to have an unacceptably high risk of death within the perioperative period despite medical optimisation. Non-operative management entailed active early mobilization without bed rest or traction. Patients managed non-operatively had a greater 30-day mortality compared with operatively managed patients. Deaths were due to pre-existing medical conditions or events, which had occurred at the time of hip fracture…” [Gregory JJ, Kostakopoulou K, Cool WP, Ford DJ. One-year outcome for elderly patients with displaced intracapsular fractures of the femoral neck managed non-operatively. Injury. 2010 Dec;41(12):1273-6. doi: 10.1016/j.injury.2010.06.009. Epub 2010 Jul 13. PMID: 20630527.]

[270]”… The pain relief achieved from surgery, could justify the risks of perioperative death, both for the patient at rest and during nursing care, even in the group at highest anesthetic risk [42]…” could be found controversial. Pain-reliefs are well accessible and give probably less dramatic side effects than operative intervention. Were there any other reasons enforcing the chosen type of therapy?

The explanation, why there was a low representation of females in the analyzed material, seems not to be persuasive. [274-8] Are women more prone to severe pulmonary insufficiency in COVID-19 infection (that is ASA 5?)?  Were there any other, not mentioned so far, limitation in the inclusion (or exclusion) into the study?

In line [283] it would be of value to cite an information about the difference of the two analyzed groups given by Pincus te al. The difference, albeit statistically significant, has been found as mild (1 vs 2%) and in a little bit different groups of patients operated by different surgeons.

In line [288] “…We observed a decrease of surgical time…” the world “lower” instead of “decrease” seems to correspond better with the meaning of the sentence.

[292-3] The sentence “…According to literature, Authors analyzed the relationship between hip prosthesis and bleeding suggesting different protocols in order to manage the patient [34]…” is hard to understand. In presented paper types of hip prostheses have not been analyzed. In the forthcoming sentence Authors explain that surgical procedures performed using both analyzed approaches were done by surgeons differing with surgical experience. Thus, are the data comparable?

[305-6] “…The mortality rate for positive COVID patients is much higher than for non-infected patients, 30-35% and 7-10% respectively [36–38]…” seems to be obvious. It supports doubts coming from relatively low mortality in the presented study (4 or 6 from 50 patients during the first 48 hours). Especially, when poor general condition (ASA III and IV) are taken into consideration.

Later on [305-6] Authors cite the publications that shows much higher mortality among COVID -positive patients.

Please change “…hypoxemic vasoconstriction…” to “hypoxic” … [329,339]. It would more specific if Authors had mentioned that this explanation (i.e. hypoxic pulmonary vasoconstriction) is in their study speculative, as they were not investigating it.

In summary, I would propose the Authors to reevaluate their Discussion focusing on their findings, which are briefly described in lines [312-20]. The Authors should not be afraid to present high mortality rate in this specific group of patients. Methinks, Authors, reducing the observation time to the first 48 postoperative hours, avoid the problematic mortality that complicates performed procedures. Unfortunately , COVID infection is complicated by high mortality rate. Nevertheless, it is an additional, independent factor that irrespective of femoral neck fracture and performed arthroplasty reduces the survival rate.

The most important problem is, whether operative treatment should be or should not be recommended in this particular group of patients. I believe that Authors, disposing of such an unique material, are able to answer this question. It might be of value in the possible, forthcoming waves of COVID infections.
